# Of Seven New K^+^ Channel Inhibitor Peptides of *Centruroides bonito*, α-KTx 2.24 Has a Picomolar Affinity for Kv1.2

**DOI:** 10.3390/toxins15080506

**Published:** 2023-08-15

**Authors:** Kashmala Shakeel, Timoteo Olamendi-Portugal, Muhammad Umair Naseem, Baltazar Becerril, Fernando Z. Zamudio, Gustavo Delgado-Prudencio, Lourival Domingos Possani, Gyorgy Panyi

**Affiliations:** 1Department of Biophysics and Cell Biology, Faculty of Medicine, Research Center for Molecular Medicine, University of Debrecen, Egyetem ter. 1, 4032 Debrecen, Hungary; shakeel.kashmala@med.unideb.hu (K.S.); umair.naseem@med.unideb.hu (M.U.N.); 2Departamento de Medicina Molecular y Bioprocesos, Instituto de Biotecnologia, Universidad Nacional Autónoma de México, Av. Universidad 2001, Cuernavaca 62210, Mexico; timoteo.olamendi@ibt.unam.mx (T.O.-P.); baltazar.becerril@ibt.unam.mx (B.B.); fernando.zamudio@ibt.unam.mx (F.Z.Z.); gustavo.delgado@ibt.unam.mx (G.D.-P.)

**Keywords:** CboK, scorpion toxin, *Centruroides bonito*, Kv1.2, Kv1.3, Kv1.2 channelopathies, electrophysiology

## Abstract

Seven new peptides denominated CboK1 to CboK7 were isolated from the venom of the Mexican scorpion *Centruroides bonito* and their primary structures were determined. The molecular weights ranged between 3760.4 Da and 4357.9 Da, containing 32 to 39 amino acid residues with three putative disulfide bridges. The comparison of amino acid sequences with known potassium scorpion toxins (KTx) and phylogenetic analysis revealed that CboK1 (α-KTx 10.5) and CboK2 (α-KTx 10.6) belong to the α-KTx 10.x subfamily, whereas CboK3 (α-KTx 2.22), CboK4 (α-KTx 2.23), CboK6 (α-KTx 2.21), and CboK7 (α-KTx 2.24) bear > 95% amino acid similarity with members of the α-KTx 2.x subfamily, and CboK5 is identical to Ce3 toxin (α-KTx 2.10). Electrophysiological assays demonstrated that except CboK1, all six other peptides blocked the Kv1.2 channel with Kd values in the picomolar range (24–763 pM) and inhibited the Kv1.3 channel with comparatively less potency (Kd values between 20–171 nM). CboK3 and CboK4 inhibited less than 10% and CboK7 inhibited about 42% of Kv1.1 currents at 100 nM concentration. Among all, CboK7 showed out-standing affinity for Kv1.2 (Kd = 24 pM), as well as high selectivity over Kv1.3 (850-fold) and Kv1.1 (~6000-fold). These characteristics of CboK7 may provide a framework for developing tools to treat Kv1.2-related channelopathies.

## 1. Introduction

Voltage-gated potassium (Kv) channels are involved in various physiological functions such as maintaining membrane potential, biasing action potential frequency, repolarizing action potential, regulating cell volume, calcium signaling, and cell proliferation in both excitable and non-excitable cells [1]. Kv channels are encoded by 40 different genes and comprised of 12 subfamilies (Kv1–Kv12), therefore being the largest ion channel family in the human genome [2,3]. Channels belonging to the Kv1 subfamily (Kv1.1–Kv1.8) are found abundantly in the nervous, vascular, and immune systems [1], Thus, pharmacological modulation of these Kv channels has significant therapeutic potential to treat cancer, autoimmune diseases, cardiovascular disorders, and neuroinflammatory diseases [4,5]. 

Kv1.2 channels, encoded by the *KCNA2* gene, are expressed majorly in the brain and spinal cord [6]. Some mutations in the *KCNA2* gene result in gain-of-function of the channel activity and lead to several channelopathies including epileptic encephalopathies [7,8] early infantile epileptic encephalopathy [9,10], and myoclonus epilepsy [11,12]. Blocking of Kv1.2 with 4-aminopyridine (a small organic compound) was shown to antagonize gain-of-function defects in KCNA2-encephalopathy patients [13]. Kv1.3 channels are predominantly present in peripheral immune cells and their upregulation in the effector memory T (T_EM_) is the key mediator of autoinflammatory diseases for example type I diabetes, rheumatoid arthritis, multiple sclerosis, and psoriasis [14,15,16,17]. Moreover, Kv1.3 ion channels also express in microglia (resident brain immune cells) and recently emerged as an attractive target to reduce neuroinflammation by modulating microglial activation [18]. For example, the inhibition of Kv1.3 by PAP-1 in multiple animal models of Parkinson’s disease leads to the decreased degeneration of dopaminergic neurons and improved behavioral outcomes [19]. Overall, since Kv1 channels are involved in a wide range of diseases, this makes Kv1 channels an attractive pharmacological target in the treatment of multiple channelopathies. 

Kv1 channels are targeted by a wide range of peptide toxins, isolated from venomous animals, which have played a significant role in identifying and developing therapeutic opportunities related to these ion channels [15,20,21]. More than 200 scorpion toxins have been identified and characterized to date which block Kv channels with high affinity with Kd values ranging from pM to nM [22]. Potassium channel scorpion toxins (KTxs) are classified into seven subfamilies based on their structural and functional characteristics (α, β, γ, δ, ε, κ, and λ) [22]. Among these, α-KTxs are the most abundantly present toxins in scorpion venom. The toxins are 23–42 amino acid residues long, and considering the similarities in their primary structure, the α-KTx family is further divided into 32 subfamilies. A structural motif known as cysteine-stabilized α/β scaffold is a characteristic of the α-KTx family, in which an α-helix and β-sheets are joined together by 3–4 disulfide bridges [23]. Moreover, a typical “functional dyad” comprised of a critically positioned Lys residue and an aromatic residue which are nine amino acids apart, is present in almost all members of the α-KTx family [24]. Upon binding of the toxin to the Kv channel, this Lys residue protrudes deep into the selectivity filter of the channel and plugs the pore, thereby inhibiting the K^+^ ion pathway [25,26,27].

The ongoing discovery of Kv channel inhibitors proves that scorpion venoms are a rich source of peptides and Mexico has a great biodiversity of different species of scorpions [28], among which at least twenty are potentially dangerous to humans [29]. In the Mexican state of Guerrero, there are a dozen different species of scorpions belonging to the genus *Centruroides*, from which a recent species, *Centruroides bonito*, was reported [30]. This manuscript describes the isolation and characterization of seven novel peptides from *C. bonito* venom. The primary structures of these peptides were determined by Edman degradation. The comparison of amino acid sequences with known potassium scorpion toxins (KTxs) and phylogenetic analysis revealed that these seven toxins belong to the α-KTx family. The peptides were characterized for their pharmacological activity on three Kv1 channels (Kv1.1–Kv1.3) by single-cell electrophysiology.

## 2. Results

### 2.1. Isolation and Sequence Determination of CboK Peptides

The three-step fractionation scheme allowed us to obtain seven peptides in pure format from the soluble venom of scorpion *C. bonito* as shown in Figure 1. The soluble venom initially separated by Sephadex G-50 produced three different fractions, named I, II, and III (Figure 1A). It is well known that fraction F-II usually contains peptides that target ion channels. Therefore, to isolate the potential ion channel blocker peptides, only fraction F-II was further separated by carboxy-methylcellulose (CMC) column. The CMC column produced 14 distinct fractions (Figure 1B). Each sub-fraction was further separated by HPLC using the C_18_ column. Most importantly, the sub-fraction 13 (F-II-13) yielded several peptides in pure format (numbered 1 to 7 in Figure 1C), which were analyzed by mass spectrometry and sequenced by Edman degradation. The MWs and amino acid sequences of all peptides are listed in Table 1. The pure peptides obtained by HPLC correspond to the following fractions, which were denominated according to the scorpion name: F-II-13.18.4 (CboK1), F-II-13.20.9 (CboK2), F-II-13.21.7 (CboK3), F-II-13.22.3 (CboK4), F-II-13.22.9 (CboK5, identical to *C. elegans* K^+^ channel toxin with a registered systematic number α-KTx 2.10), F-II-13.23.2 (CboK6), and F-II-13.25.3 (CboK7) [31]. Reduced and carboxymethylated forms of the peptides CboK1, CboK2, CboK3, CboK5, and CboK6 were subjected to Edman degradation and the complete amino acid sequences were obtained in a direct mode. Direct sequencing of peptide CboK4 allowed the identification of the N-terminal residues from Thr1 to Met30. The endopeptidase V8 digested peptide produced a sub-peptide (eluted at time 18.9 min from the HPLC) that allowed the identification of C-terminal residues from Glu27 to the last residue Pro39. A similar procedure with peptide CboK7 allowed the direct identification of the first 22 amino acid residues, from Thr1 to Gly22. After enzymatic digestion with endopeptidase V8, a sub-peptide eluted from HPLC at 24.9 min allowed the identification of Glu19 to the last residue Val39.

### 2.2. Phylogenetic and Sequence Analysis of C. bonito KTx Toxins

Sequence analysis showed that *C. bonito* toxins can be assigned to three groups with different lengths in the primary sequence. The first group is composed of toxins with lengths of 32 to 33 amino acids. The toxins Cobatoxin-1 (α-KTx 10.1; O46028), Cobatoxin-2 (α-KTx 10.2; P58504) from *C. noxius*, and Toxin II.10.4 (α-KTx 10.4; C0HJW2) from *C. tecomanus* share between 84 and 97% amino acid identity with CboK2 (α-KTx 10.6; C0HM75), while CboK1 (α-KTx 10.5; C0HM73) shares only 64% sequence identity with CboK2 (Figure 2A). In the second group, which consists of peptide toxins with a length of 38 amino acids, toxin CboK5 (α-KTx 2.10; C0HM74) and Ce3 from *C. elegans* (α-KTx 2.10; P0C163) share 100% amino acid identity, while CboK5 and CboK6 (α-KTx 2.21; C0HM76) share 95% identity. Other related toxins are CllTx1 from *C. limpidus* (α-KTx 2.3; P45629), Toxin II.10.5 from *C. tecomanus* (α-KTx 2.15; C0HJW1), and Css20 from *C. suffusus* (α-KTx 2.13; P85529) share amino acid sequence identity percentages between 84 and 92% with CboK5 (Figure B). The third group is formed by toxins with a length of 39 amino acids, toxins CboK3 (α-KTx 2.22; C0HM77), CboK4 (α-KTx 2.23; C0HM72), CboK7 (α-KTx 2.24; C0HM78), Toxin II.12.5 (α-KTx 2.16; C0HJW6), and Ct28 from *C. tecomanus* (α-KTx 2.20; P0DUI4) share between 90 and 97% amino acid sequence identity (Figure 2C).

Phylogenetic analysis of *C. bonito* toxins and other alpha family potassium toxins (α-KTx) showed that the toxins cluster into two strongly supported major clades (percent posterior probability of 100) (Figure 3). In the first clade, there are the α-KTx toxins with short sequences corresponding to subfamily 10, the toxins CboK1 (α-KTx 10.5) and CboK2 (α-KTx 10.6) from *C. bonito* join the other three toxins from two previously reported species of the genus *Centruroides* (Cobatoxin-1, Cobatoxin-2, and Toxin II-10.4) [32,33].

In the second major clade of the α-KTx, there are toxins of subfamilies 2 and 4. The toxins CboK5 (α-KTx 2.10) and CboK6 (α-KTx 2.21) are part of a poorly supported basal subclade (percent posterior probability less than 50) with other toxins from subfamily 2. In this clade, Ce3 from *C. elegans* is also present, with which CboK5 shares 100% amino acidic identity corresponding to the mature chain [34]. Other phylogenetically related toxins in this clade are CllTx1, Toxin II-10.5, and Css20 [6,33,35]. The toxins CboK3 (α-KTx 2.22), CboK4 (α-KTx 2.23), and CboK7 (α-KTx 2.24) are in a phylogenetically more recent clade compared to CboK5 and CboK6. These toxins from *C. bonito* are the longest in length (39 residues) and are phylogenetically related to the Toxin II-12.5 from *C. tecomanus* [33]; in the earlier and the later clades, other members of subfamily 2 are found.

### 2.3. Pharmacological Characterization of CboK Peptides 

The sequence analysis and phylogenetic tree reveal that the seven CboK peptide toxins bear a great deal of resemblance to the α-KTxs that block voltage-gated potassium (Kv1) channels. Based on the similarity index of CboK peptides with other α-KTxs and the literature available for the activity of closely related peptides [36,37], Kv1.1, Kv1.2, and Kv1.3 K^+^ channels were selected as the potential target of CboK peptides. Macroscopic Kv1.1 and Kv1.2 currents were measured in transiently transfected CHO cells. For Kv1.3 current recording, human T lymphocytes were activated by Phytohemagglutinin A (PHA) to increase Kv1.3 channel expression, and micro-pipettes were filled with Ca^2+^-free solution to prevent KCa3.1 channel activation. K^+^ currents were evoked by applying 15-ms-long depolarization pulses to +50 mV from a holding potential of −120 mV except for Kv1.2, as it has highly variable activation kinetics; therefore, 15–500 ms-long pulses were applied to maximize the open probability of Kv1.2 channel [38]. CboK peptide toxins were dissolved in a freshly prepared extracellular solution and applied to the whole-cell patch-clamped cells using a micro perfusion system at 200 μL/min flow rate. The perfusion system’s flow rate and complete solution interchange were assessed frequently using fully reversible blockers as positive controls at a concentration equivalent to their Kd values, i.e., 0.3 mM tetraethylammonium (TEA) for Kv1.1 (Figure 4A), 14 nM Charybdotoxin (ChTx) for Kv1.2 (Figure 4B), and 10 mM TEA for Kv1.3 (Figure 4C). Approximately 50% reduction in K^+^ current at equilibrium block while perfusing the cells with positive controls indicated the proper operation of the perfusion system and confirmed the nature of the expressed current.

Figure 4A–C shows the representative whole-cell current traces for Kv1.1 (Figure 4A), Kv1.2 (Figure 4B), and Kv1.3 (Figure 4C) channels, recorded sequentially in the same cell for each channel, in the presence of control solution (black trace) and at equilibrium block while perfusing the cell with positive control (red trace) or the CboK7 toxin at concentration of 100 nM in case of Kv1.1 and Kv1.3 and 1 nM for Kv1.2 (purple trace). At steady-state block, CboK7 inhibited ~42% of Kv1.1 and ~80% of Kv1.3 currents at 100 nM and >99% of Kv1.2 currents at 1 nM concentration. Figure 4D–F represents the summarized pharmacological screening data of CboK1 to CboK7 toxins for Kv1 channels. Generally, all the peptides were tested at 100 nM concentration except CboK2 to CboK7 peptides in case of Kv1.2, where 1 nM concentration was used for screening due to high affinity of these peptides for Kv1.2.

We found that CboK3 and CboK4 slightly inhibited (<10%) the Kv1.1 currents at 100 nM concentration; however, CboK7 showed ~42% block of Kv1.1 at this concentration (Figure 4D). The low potency of block and limited availability of CboK peptides from the native source restricted us from further studying the concentration-dependent block of the Kv1.1 channel. Figure 4E shows that the peptides CboK2 to CboK6 at 1 nM concentration reduced the Kv1.2 currents by 60−92%, whereas CboK7 showed outstanding potency for Kv1.2 by inhibiting ~96% of the current at 1 nM. On the other hand, Kv1.2 was totally insensitive to 100 nM of CboK1 toxin. In the case of Kv1.3, CboK2–CboK7 peptides demonstrated moderate inhibitory effects with 41–81% block at 100 nM, whereas CboK1 showed less than 4% inhibition.

### 2.4. CboK Peptides Inhibit Kv1.2 with Picomolar Affinity

All CboK peptides except CboK1 exhibited remarkable blocking effects on Kv1.2 currents when tested at 1 nM concentration. This motivated a more detailed study to better characterize the activities of CboK peptides on the Kv1.2 channel. Figure 5A shows the representative Kv1.2 current traces before and after application of 1 nM CboK5 which inhibited ~75% of the peak current. The onset and recovery from the block of Kv1.2 currents at 1 nM concentration of CboK5 are shown in Figure 5B. Normalized peak currents (I_norm_ = I_t_/I_0_, where I_t_ is the peak current in the presence of the toxin at time t and I_0_ is the peak current in the control solution at t = 0), were plotted as a function of time. The block was completely reversed by applying a toxin-free control solution to the cell. The kinetics of both association and dissociation of all the CboK peptides were slow as demonstrated for CboK5 peptide as an example (Figure 5B), it took several minutes to achieve equilibrium block and to recover completely the peak current measured prior to toxin application. Consequently, slow toxin binding kinetics led us to generate the concentration–response curves for CboK peptides in a cumulative manner. 

The concentration–response relationships of CboK2 to CboK6 for Kv1.2 channel inhibition are shown in Figure 5C. Different concentrations of each peptide were applied to the cells for a sufficient period to achieve the equilibrium block, considering the slow blocking kinetics, especially at low toxin concentrations. The RCF values were calculated, and data points were fitted with the Hill equation (see Materials and Methods for specifics) to obtain concentration–response curves. The resulting dissociation constants (Kd) and the Hill coefficients were Kd = 763 pM and H = 1.5 for CboK2, Kd = 106 pM, and H = 1.2 for CboK3, Kd = 125 pM and H = 1.0 for CboK4, Kd = 376 pM and H = 1.2 for CboK5, Kd = 585 pM and H=1.3 for CboK6, and Kd = 24 pM and H = 1.3 for CboK7, respectively (Figure 5C). Out of seven CboK peptides, CboK7 is outstanding with a high affinity of 24 pM for Kv1.2. 

### 2.5. CboK Peptides Display Nanomolar Affinity for Kv1.3

CboK peptides also target the Kv1.3 ion channel as described in the initial screening experiments earlier (Figure 4). Therefore, we investigated the effect of these peptides in a concertation-dependent manner on Kv1.3. The affinity of CboK1 for Kv1.3 was quite low in the screening and due to limited supply of venom, the full concentration–response relationship was not obtained. Data were not obtained for CboK5 either, as it is 100% identical to a previously known α-KTx 2.10; Toxin Ce3, which was already characterized for Kv1.3 [34]. Figure 6A shows the whole-cell Kv1.3 current traces recorded chronologically on the same cell in the presence of control solution (black trace) and after reaching the equilibrium block by applying 10 mM TEA (as positive control, red trace) or in the presence of 100 nM of CboK3 (blue trace). CboK3 at 100 nM concentration inhibited ~80% of the current. The development of steady-state block and recovery of the initial current at various concentrations of CboK3 toxin is presented in Figure 6B. The Kv1.3 block by CboK3 and all the other CboK toxins was fully reversible after perfusing the cell with the toxin-free external solution and had much faster binding kinetics than for Kv1.2. The onset of the steady-state block at given concentration of CboK3 (5, 15, 40, 100, 350 nM) and complete recovery from the block took place in the course of ~1 min. 

However, while studying the inhibition of Kv1.3 currents by CboK peptides, we observed an interesting phenomenon, i.e., the whole-cell current showed a steady increase during the course of the 15-ms-long depolarization in the presence of the peptides, which was more obvious at high (>50 nM) peptide concentrations (Figure 6A,C). Therefore, the remaining current fractions, determined from the current amplitudes at the end of the depolarizations were larger than at earlier time points, e.g., when the control traces reached the peak (Figure 6A, vertical dashed line). Based on the very fast off rate of the peptides, we hypothesized that the steady rise in the current (Figure 6A,C) may be the consequence of unbinding of the peptides, induced by the depolarized test potentials and the permeating potassium ions, known as knock-off [27,34,39]. To confirm this phenomenon, we compared the effect of the CboK2 peptide at 1 µM (sufficient native toxin material was available for this peptide) at two different test potentials i.e., +50 mV and 0 mV applied for 15 ms (Figure 6C,E). There are two important differences in the currents between Figure 6C,E measured in the presence of CboK2. First, the steady increase in the current in the presence of CboK2 is absent in Figure 6E, and second, the RCF measured at the end of the pulse in 1 µM CboK2 was ~0.34 at +50 mV (Figure 6C), whereas the RCF value was ~0.23 at 0 mV (Figure 6E). The statistical analysis showed that the RCF is smaller at 0 mV test potential as compared to the one determined at +50 mV test potential (Figure 6G). Another indication of the dissociation of the peptide from Kv1.3 at depolarized test potentials is shown in Figure 6D when 1000 ms depolarizing test pulses were applied to +50 mV. The Kv1.3 current shows a characteristic inactivation kinetics in the absence of the peptides (black) whereas the kinetics is drastically altered in the presence of 1 µM CboK2 (blue): (1) there is an additional, very slow component of the activation kinetics of the current, (2) the absence of the sharp peak of the current, (3) the slow and incomplete inactivation kinetics, and (4) the crossover of the current traces recorded in control solution or in the presence of CboK2. These characteristic changes in the shape of the Kv1.3 current are consistent with time-dependent unbinding of the CboK2 during the depolarization. Repeating the experiments in Figure 6D at 0 mV test potential (Figure 6F) shows that the unbinding of CboK2 is reduced (i.e., more block), and the very slow component of current activation is virtually missing. On the other hand, the changes in the inactivation kinetics are still observable (slow and incomplete inactivation and crossover).

The data above suggest that at +50 mV test potential, significant unbinding of CboK toxins takes place, and therefore 0 mV test potential should be used to determine the RCF values accurately from the peak currents. However, due to the insufficient quantity of native peptides, we were unable to repeat the experiments at 0 mV test potential, where peak currents are reliably determined in the absence of significant unbinding. Therefore, for concentration–response curves, current amplitudes were determined at the isochronal time point of 4 ms as indicated with vertical dashed line in Figure 6A, where the control traces would display the peak current. Isochronal RCF data points obtained this way (isochronal RCF = I_4ms_/I_4ms,0_ where I_4ms,0_ and I_4ms_ are the currents at 4 ms time point in the absence and presence of CboK peptides, respectively) at different toxin concentrations were fitted with the Hill equation to obtain concentration–response curves as shown in Figure 6H. The resulting Kd values and the Hill coefficients were Kd = 171 nM and H = 0.8 for CboK2, Kd = 34.3 nM, and H = 0.96 for CboK3, Kd = 21.7 nM and H = 0.94 for CboK4, Kd = 160 nM, and H = 0.77 for CboK6 and, Kd = 20.4 nM and H = 0.8 for CboK7, respectively.

## 3. Discussion

The city of Acapulco in the state of Guerrero, Mexico, has a newly described species of *Centruroides* scorpion, named *Centruroides bonito* [30], which is dangerous to humans [29] and its venom composition is not known. In this article, we are reporting for the first time the isolation and chemical and physiological characterization of seven novel peptides isolated from the venom of this scorpion. The separation of these peptides was obtained by applying the soluble venom through the normal procedure used in our laboratory for this purpose, as described, using gel filtration on Sephadex G-50 column, followed by ion-exchange separation in carboxy-methyl cellulose resins and finally by high-performance liquid chromatography. The primary structure of all these peptides was obtained by Edman degradation and confirmed by mass spectrometry. Comparative analysis of CboK toxins amino acid sequences showed high similarities with other known peptide toxins affecting K^+^ conduction through ion channels. They belong to the α-KTx family of K^+^-channels-specific peptides. Phylogenetic tree analysis revealed that Cbok1 and Cbok2 toxins are new members of α-KTx family 10 with systematic name, α-KTx 10.5 and α-KTx 10.6, respectively. Cbok3 to CboK7 toxins are classified as the new members of α-KTx family 2 with systematic names α-KTx 2.22, α-KTx 2.23, α-KTx 2.10, α-KTx 2.21, and α-KTx 2.24, respectively.

Based on the similarities in primary structure of CboK peptides with the previously known scorpion toxins, we identified the Kv1.1, Kv1.2, and Kv1.3 as the potential targets to characterize the pharmacological activities of CboK peptides. Moreover, all the CboK peptides possess the typical functional dyad of Lys and Tyr, which is considered essential for high affinity block [25,26]. The blocking potencies (Kd values) of close relatives of each CboK peptide are listed in Table 2. 

Although CboK1 peptide showed 68% and 66% sequence homology with cobatoxin (CoTx1, α-KTx 10.1) isolated from *Centruroides noxius* [32,40] and peptide II-10.4 (α-KTx 10.4) isolated from *Centruroides tecomanus* [33], which inhibit Kv1 channels with moderate affinity (Kd values ranging between nM to µM), it did not show any effect on Kv1.1–Kv1.3 channels at 100 nM concentration. Moreover, CboK1 has a cysteine pattern similar to many of the α-KTxs peptides and also has the Lys–Tyr dyad.

CboK2 peptide inhibited Kv1.2 with high affinity (Kd = 760 pM) and showed less affinity for Kv1.3 with Kd = 171 nM, whereas it did not block Kv1.1 at 100 nM. The amino acid sequence of CboK2 is 97% identical to CoTx1 (α-KTx 10.1, isolated from *Centruroides noxius*) [40], having a difference of just one amino acid: Thr1 instead of Ala1. CoTx1 inhibits Kv1.2 with Kd = 27 nM but has very low affinity for Kv1.1 and Kv1.3. CboK2 is also highly similar (94%) to Toxin II-10.4 (α-KTx 10.4, isolated from *Centruroides tecomanus*) [33] and CoTx2 (84%) (α-KTx 10.2, from *Centruroides noxius*) [32]. Toxin-II 10.4 blocks Kv1.2 and Kv1.3 but has no effect on Kv1.1. On the other hand, CoTx2 mildly inhibits only Kv1.1 (Kd = 1 µM) but does block Kv1.2 and Kv1.3.

CboK3, CboK4, and Cbok7 toxins are 39 amino acids residues long and have quite identical (>97% similarity) primary structures with each other. CboK3 differs from CboK4 by two amino acids, having Ile2 instead of Phe2 and Val39 instead of Pro39. However, this difference does not drastically impart any change in its affinity for Kv1.1, Kv1.2, and Kv1.3. CboK3 and Cbok4 blocks Kv1.2 with high affinity (106 and 125 pM, respectively), Kv1.3 with moderate affinity (34.3 and 21.7 nM, respectively) and Kv1.1 with low affinity (1 µM and 1.5 µM, respectively). In contrast, CboK7 differs from CboK3 and CboK4 by only a single amino acid but at different positions. CboK7 has Phe at position 2 instead of Ile in CboK3 and Val at position 39 instead of Pro in CboK4. Interestingly, it inhibits Kv1.2 with significantly high affinity Kd = 24 pM and moderately blocks Kv1.1 and Kv1.3 with 141 nM and 20.4 nM Kd values. This shows that the presence of Phe2 and Val39 greatly increases the binding of CboK7 to Kv1.1 and Kv1.2 ion channels. In comparison with the previously described peptides, CboK3, CboK4, and CboK7 are >95% identical to Toxin II-12.5 (α-KTx 2.16, isolated from *Centruroides tecomanus*), ~85% to Ce4 toxin (α-KTx 2.11, *Centruroides elegans*) [34], and ~80% to Toxin II-12.8 (α-KTx 2.17, *Centruroides tecomanus*). Toxin II-12.5 inhibits Kv1.2 (Kd = 0.7 nM) and Kv1.3 (Kd= 26.2 nM) channels but has no effect on Kv1.1 at 10 nM and Toxin II-12.8 inhibits Kv1.1 (Kd = 4.8 nM) and Kv1.2 (Kd = 2.9 nM) but has no effect on Kv1.3 at 10 nM concentration [33]. Ce4 was shown to inhibit Kv1.3 (Kd = 0.98 nM) but was not assessed for Kv1.1 and Kv1.2 blocking potencies [34]. 

CboK5 is 100% identical with a previously reported toxin Ce3 (α-KTx 2.10) isolated from another scorpion *Centruroides elegans*. CboK5 peptide blocked Kv1.2 with Kd value of 376 pM and Kv1.3 with Kd = ~126 nM, and did not inhibit Kv1.1 currents at 100 nM concentration. On the other hand, Ce3 inhibits Kv1.3 with an estimated Kd = 366 nM and its effect on Kv1.1 and Kv1.2 is unknown [34]. CboK6 peptide has a sequence similar to CboK5 and Ce3 (95%), Toxin II-10.9 (92%) (α-KTx 2.3), and Toxin II-10.5 (84%) (α-KTx 2.15). CboK6 blocks Kv1.2 and Kv1.3 with Kd values of 585 pM and 160 nM, respectively, and does not block Kv1.1. Toxin II-10.9 is not active on any of the three ion channels (Kv1.1–Kv1.3) at 10 nM and Toxin II-10.5 also does not affect Kv1.1 but inhibits Kv1.2 (0.3 nM) and Kv1.3 (8.3 nM) [33].

Several peptide toxins have been identified and characterized that inhibit K^+^ channels with high affinity ranging from pM to nM concentration. However, in general, native toxins target more than one ion channel which compromises their therapeutic application, for example, Margatoxin (α-KTx 2.22), Urotoxin (α-KTx 6.21), Cm39 (α-KTx 4.8), etc. [41,42,43,44]. Only a few toxins are known which have high affinity and reasonable selectivity for Kv1.2 over other closely related ion channels. For examples, Maurotoxin (α-KTx 6.2) inhibits Kv1.2 (Kd = 0.8 nM) with ~56-fold selectivity over Kv1.1 and 225-fold over Kv1.3 [45,46]. CoTx1 (α-KTx 10.1) has moderate affinity for Kv1.2 (27 nM) but exhibits ~900-fold selectivity over Kv1.1 and ~200-fold over Kv1.3. It also inhibits KCa2.x and KCa3.1 channels [40]. Pi4 (α-KTx 6.4), isolated from *Pandinus imperator* scorpion, is the most potent inhibitor of Kv1.2 with Kd = 8 pM and highly selective (>1.25 million times) over Kv1.1 and Kv1.3. However, it also blocks KCa2.x (500 nM) and Shaker K^+^ channels (3 pM) [47,48]. 

Among the CboK toxins reported in this article, CboK7 is a remarkable inhibitor of Kv1.2 (Kd = 24 pM) with great selectivity of ~6000-fold over Kv1.1 and 850-fold over Kv1.3. In order to increase the selectivity profile of natural toxins with therapeutic potential, protein engineering could be used; for example, an engineered analog of Anuroctoxin (α-KTx-6.12) with double substitution (N17A/F32T) was developed which retained its natural potency for Kv1.3, while gaining 16,000-fold selectivity over Kv1.2 [49]. The selectivity of CboK7 toxin for a Kv1.2 over Kv1.3 can be further improved through peptide engineering after uncovering the residues which interact with the ion channel vestibule. Nevertheless, the native selectivity profile makes Cbok7 an outstanding candidate for selective inhibition of Kv1.2 and it can be further exploited to treat gain-of-function associated Kv1.2 channelopathies. Kv1.2 ion channels are majorly present in the central nervous system and gain of function mutations have been identified in patients with epileptic encephalopathy [8]. Thus inhibition of Kv1.2 may have beneficial effects in epilepsy associated with gain-of-function mutations [12]. Moreover, since Kv1.2 is present in the brain and is protected by the blood–brain barrier (BBB), which has a restrictive nature, it hinders the drugs, including peptide toxins, from reaching the ion channels in the brain tissues [50]. This issue can be overcome by conjugating the therapeutic toxins to BBB-targeting shuttle peptides [51,52,53] which will carry them to the brain tissues and target the Kv1.2 channels.

## 4. Conclusions

In conclusion, we describe seven new peptide toxins in the venom from recently reported Mexican scorpion *Centruroides bonito*. These peptides are composed of 33–39 amino acid residues, held together by cross-linking of three potential disulfide bonds, which inhibits the Kv1.2 channel at picomolar concentration and affects the normal function of Kv1.1 and Kv1.3 at nanomolar concentration except CboK1 peptide (33 residues long) which is relatively less effective on these ion channels. All the CboK peptides belong to the previously classified α-KTx family of scorpion toxins. The pharmacological properties of the CboK7 peptide among these new toxins are outstanding which makes this peptide an attractive therapeutic candidate for gain-of-function associated Kv1.2 channelopathies. 

## 5. Materials and Methods

### 5.1. Venom Source and Purification Procedure

The scorpions used in these experiments were collected in the State of Guerrero (SEMARNAT permission SGPA/DGVS/00367/22). The venom obtained by electric stimulation was suspended in water and submitted to centrifugation at 15,000× *g* for 15 min. The soluble part of the venom was lyophilized and kept at −20 °C until used. 

The initial separation of the venom (~80 mg protein content) was obtained by gel filtration using a Sephadex G-50 column and the corresponding fractions were applied to an ion-exchange column prepared with carboxy-methylcellulose (CMC) beads as described elsewhere [33]. The subfraction F-II-13 was further purified by high-performance liquid chromatography (HPLC) using a Waters model 1525 chromatographer (Milford, MA, USA). An analytical C_18_ reverse-phase column from Vydac (Hisperia, CA, USA) was used for the separation of F-II.13, using the same conditions, as earlier described [33]. The pure components obtained were analyzed by mass spectrometry and the primary structure was determined by automatic Edman degradation as described in Section 2.2.

### 5.2. Mass Spectrometry and Sequence Determination

The molecular weight (MW) of the purified peptides was determined using an LCQFleet mass spectrometer from Thermo Fisher Scientific Inc. (San Jose, CA, USA). The full primary structure of the purified peptides was obtained by Edman degradation, using a Shimadzu Protein Sequencer PPSQ-31A/33A (Columbia, MD, USA). For this purpose, two procedures were used: direct sequencing of native peptides and second, sequencing after reduction and alkylation with iodoacetic acid for the correct determination of the cysteine residues. Only two purified peptides (CboK4 and CboK7) required enzymatic digestion of the reduced and alkylated peptide, and further separation by HPLC to complete the full sequence (see Olamendi-Portugal et al. [33]). The two reduced and carboxylated peptides (CboK4 and CboK7) were treated with Glu-C endopeptidase v8 purchased from the Roche Diagnostics GmbH (Mannheim, Germany). The peptides dissolved in 25 mM ammonium bicarbonate buffer pH 8.4 were digested for 4.5 h and separated by HPLC. The sub-peptides obtained from this HPLC separation were sequenced, resulting in the determination of the overlapping sequences shown in Table 1. The molecular weights obtained were confirmed by mass spectrometry. The amino acid sequences of all other peptides (CboK1, CboK2, CboK3, CboK5, CboK6) were obtained directly, without the need for enzymatic digestion. The amino acids found, and the expected molecular masses obtained, confirmed the sequences.

### 5.3. Phylogenetic Analysis and Systematic Name Assignment of New Scorpion Toxins

The search for phylogenetically related toxins was performed with BLASTP v2.13.0+ [54] using the 198 scorpion potassium toxins (KScTx) listed in kaliumdb (Database of polypeptide ligands of potassium channels database) [22] as a database for the query. An e-value of 1 × 10^−10^ was set as a significant cut-off. Sequences of mature chains were aligned using the E-INS-i iterative refinement method of MAFFT v7.490 [55]. Phylogenetic tree construction by Bayesian inference was performed with MrBayes v3.2.7 [56] with the options lset rates = gamma with prset aamodelpr = fixed. The best substitution model for phylogenetic analysis (WAG) was estimated using ModelFinder [57,58]. The analysis was run for 1 × 10^7^ generations in eight chains with a sampling frequency of every 1000 trees. The tree was edited using FigTree v1.4.4 and Affinity Designer v1.8.5.703. Systematic name assignment of the potassium toxins of *C. bonito* was performed as proposed by [59]. The current numbering of each potassium toxin subfamily was consulted on the VenomZone and kaliumdb websites (https://venomzone.expasy.org/3438 and https://kaliumdb.org; both accessed on 1 August 2022).

### 5.4. Isolation and Activation of PBMCs

Human peripheral venous blood was acquired from anonymous healthy volunteers from the blood bank. Blood samples were collected by having approval from the Ethical Committee of the Hungarian Medical Research Council (36255-6/2017/EKU). PBMCs were separated by Histopaque 1077 (Sigma-Aldrich, Budapest, Hungary) separation technique, and isolated cells, at the density of 5
×
10^5^ cells, were cultured in RPMI 1640 media (Gibco, Grand Island, NY, USA) supplemented with 10% FBS (Sigma-Aldrich, Budapest, Hungary), 2 mM L-glutamine, 100 mg/mL streptomycin, and 100 mg/mL penicillin in a humidified incubator at 37 °C and 5% CO_2_ for 3–6 days. A measure of 2, 5, or 10 μg/mL of Phytohemagglutinin A (PHA) (Sigma-Aldrich, Budapest, Hungary) was added in cultured cells to activate the PBMCs and upregulate the Kv1.3 expression.

### 5.5. Heterologous Expression of Kv1.1 and Kv1.2 Channels in CHO Cells

Chinese hamster ovary (CHO) cells (gift from Yosef Yarden, Weizmann Institute of Science, Rehovot, Israel) were grown in Dulbecco’s modified Eagle’s medium (DMEM, cat# 11965084; Gibco, Grand Island, NY, USA) containing 100 μg/mL streptomycin, 100 U/mL penicillin-g 2 mM L-glutamine and 10% FBS, in an incubator at 37 °C and 5% CO_2_. Cell passage was done twice a week following incubation in 0.05% trypsin–EDTA solution for 5 min. 

CHO cells were transiently transfected by using Lipofectamine 2000 kit (Invitrogen, Carlsbad, CA, USA), as per the manufacturer’s protocol with vector pCMV6-AC-GFP (cat# RG211000 and RC222200; OriGene Technologies, Rockville, MD, USA) encoding hKv1.1 (hKCNA1 gene) or hKv1.2 (hKCNA2 gene) ion channels. Cells were grown under standard conditions in an incubator at 37 °C and 5% CO_2_ for 24 h. GFP-positive transfected cells were identified by utilizing a Nikon TMS fluorescence microscope (Nikon, Tokyo, Japan) using bandpass filters of 455–495 nm and 515–555 nm for excitation and emission, respectively. Of the GFP-positive cells, 60−70% expressed the transfected ion channels.

### 5.6. Electrophysiology

The patch-clamp technique was used for measuring whole-cell currents in voltage-clamped mode using standard protocols [60]. Electrophysiological recordings were acquired by using Multiclamp 700B amplifier connected to a personnel computer with an Axon Digidata1440 digitizer (Molecular Devices, Sunnyvale, CA, USA). Pipettes were pulled from GC150F-15 borosilicate capillaries (Harvard Apparatus, Kent, UK) having 3−5 MΩ tip resistance in the bath solution. All current recordings were carried out at room temperature (20−25 °C) and only those currents were recorded and subjected to analyses when the leak current at holding potential was <5% of peak current at the test potential. Current traces were lowpass-filtered by utilizing the built-in analog four-pole Bessel filters of the amplifiers and were sampled at 20 kHz which is at least twice the filter cut-off frequency. A gravity-driven perfusion system was used to perfuse the cells with control and test solutions and excess fluid was removed continuously through vacuum suction. The extracellular solution (bath solution) contained 145 mM NaCl, 5.5 mM glucose, 5 mM KCl, 2.5 mM CaCl_2_, 1 mM MgCl_2_, and 10 mM HEPES, having pH of 7.35 and an osmolarity between 302 and 308 mOsM/L. Toxins were dissolved in bath solution at different molar concentrations and supplemented with 0.1 mg/mL of bovine serum albumin BSA (Sigma-Aldrich, Budapest, Hungary). The intracellular solution (pipette filling) was comprised of 140 mM KF, 2 mM MgCl_2_, 1 mM CaCl_2_, 10 mM HEPES, and 11 mM EGTA, with a pH of 7.22 and an osmolarity of ~295 mOsM/L.

Generally, to evoke the Kv1.xcurrents, 15-ms-long depolarization pulses to +50 mV from a holding potential of −120 mV were applied, and the current traces were recorded after every 15 s. However, in the case of Kv1.2, due to its variable activation kinetics, the duration of the depolarization pulse was set between 15 and 500 ms to attain the saturated peak current. The Clampfit 10.7 software package (Molecular Devices, Sunnyvale, CA, USA) was used to analyze the current records and before analysis all the current traces were digitally filtered (three-point boxcar smoothing). The blocking effect of the toxin at a given concentration was determined as the remaining current fraction (RCF = I/I_0_, where I is the peak current (but see Figure 6) at equilibrium block at a given toxin concentration or peak current after perfusing the cells with toxin solution for 3–5 min in case of no block and I_0_ is the peak current in the absence of the toxin). The data points in the concentration-response curve represent the average of three to five individual cells and error bars indicate the standard error of the mean (SEM). Data points were fitted with the Hill equation: RCF = Kd^H^/(Kd^H^ + [toxin]^H^), where Kd is the dissociation constant, [toxin] is the concentration of the toxin and, H is the Hill coefficient.

### 5.7. Statistics

For graph plotting and statistical analysis, GraphPad Prism software (version 8.0.1, La Jolla, CA, USA) was used. For pairwise comparison, Student’s *t*-test was used and for multiple comparisons, one-way ANOVA with post-hoc Dunnet’s test was performed. Data were presented with standard error of mean (SEM).

## Figures and Tables

**Figure 1 toxins-15-00506-f001:**
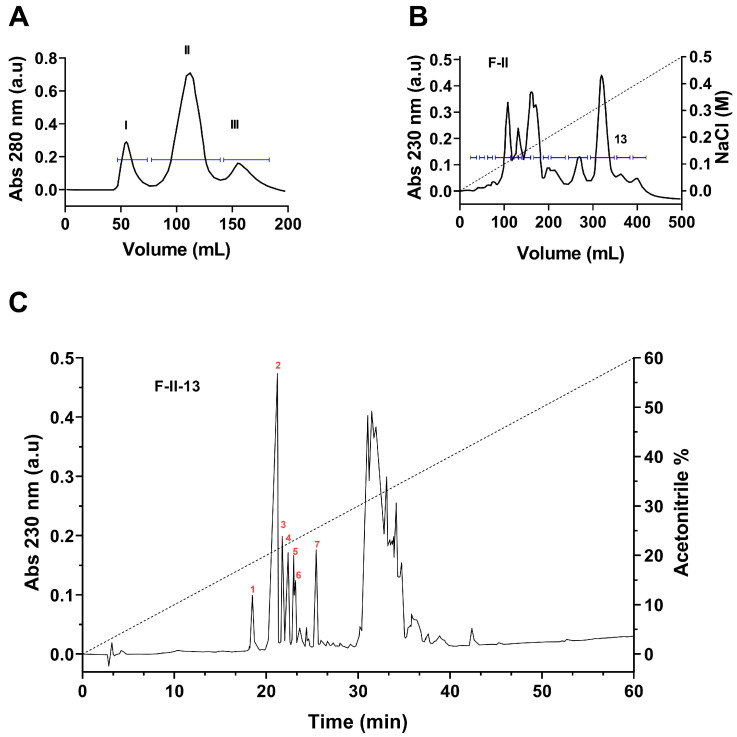
Separation of peptide toxins from the venom of *Centrurioides bonito*. (**A**) Fractionation of soluble venom by means of gel filtration (Sephadex G-50). Three main fractions were separated (I, II, and III). (**B**) Fraction II, containing the toxins was further separated by ion exchange chromatography (CMC), obtaining 14 fractions. (**C**) Subfraction 13 was applied to a C_18_ HPLC column which made it possible to obtain several pure peptides (numbered 1 to 7).

**Figure 2 toxins-15-00506-f002:**
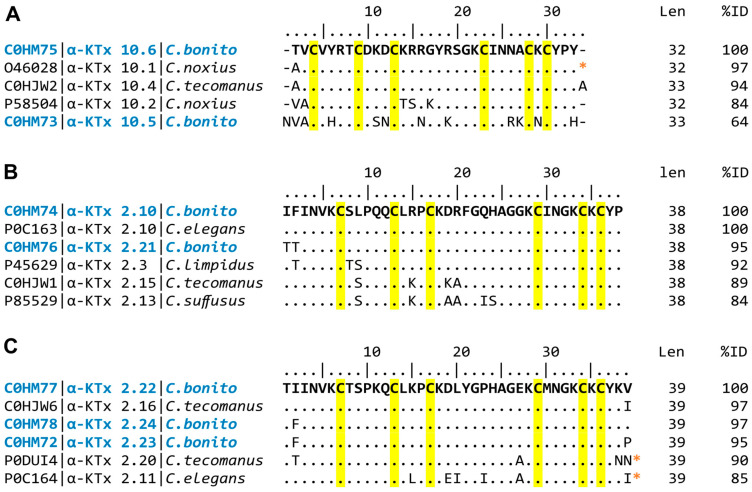
Multiple alignments of *C. bonito* toxins (text in blue) and other scorpion potassium toxins. KaliumDB sequences closely related to (**A**) CboK1 (α-KTx 10.5) and CboK2 (α-KTx 10.6); (**B**) CboK5 (α-KTx 2.10) and CboK6 (α-KTx 2.21); (**C**) CboK3 (α-KTx 2.22), CboK4 (α-KTx 2.23), and CboK7 (α-KTx 2.24) are shown. The UniProt access code is shown on the left, followed by the systematic name and species name. Len indicates mature chain length; %ID indicates percent amino acid identity. Conserved cysteine residues are highlighted in yellow. Identical positions to *C. bonito* toxins are indicated by dots. (*) indicates that the C-terminal is amidated.

**Figure 3 toxins-15-00506-f003:**
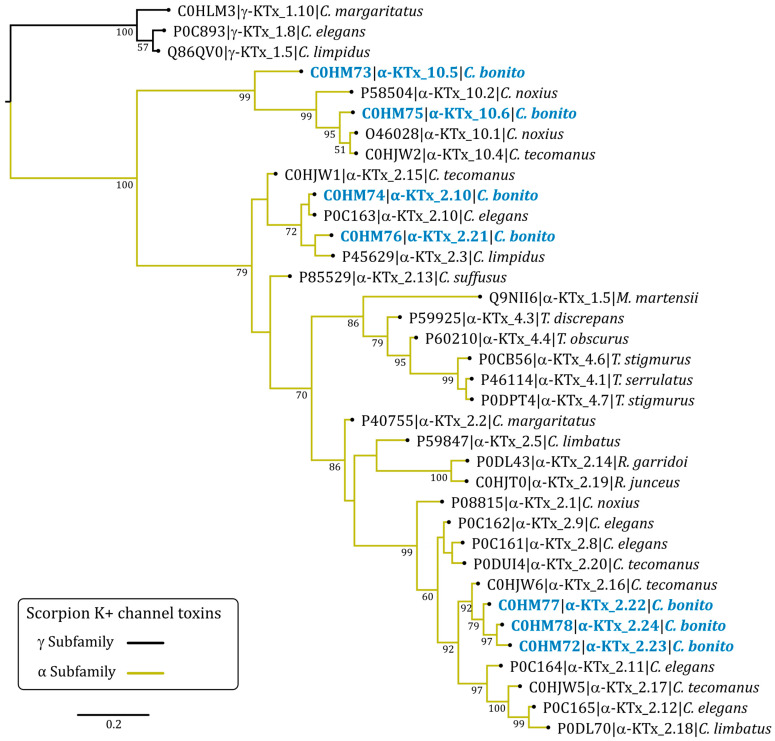
Bayesian phylogenetic tree of *C. bonito* toxins and other potassium toxins (KTx) from scorpion. Numbers below the nodes indicate percentage posterior probability values greater than 50. Sequence names are composed of the UniProt accession number, followed by the systematic name of the toxin and the species name. Three γ-KTx (UniProt accession no. P0C893, Q86QV0, and C0HLM3) were used as outgroups.

**Figure 4 toxins-15-00506-f004:**
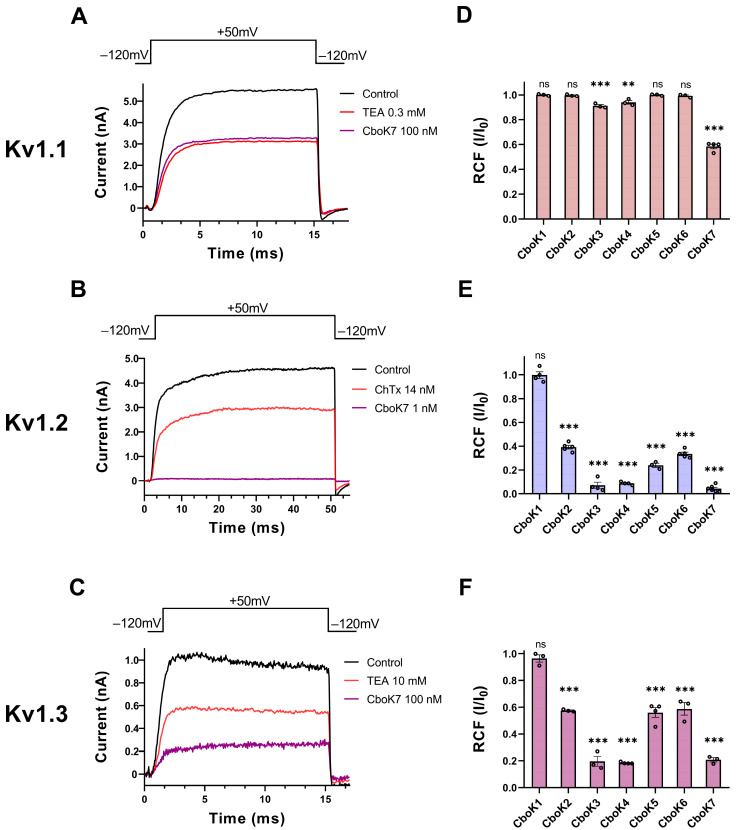
Inhibition of Kv1 channels by CboK7. (**A**–**C**) Whole-cell Kv1.1 (**A**) and Kv1.2 (**B**) currents were recorded in transiently transfected CHO cells whereas Kv1.3 currents (**C**) were measured in an activated human peripheral T lymphocyte. Here, 15-ms (for Kv1.1 and Kv1.3) or 50-ms-long (Kv1.2) voltage pulses were applied to +50 mV from a holding potential of −120 mV and current traces were recorded every 15 s. Representative traces show the K^+^ currents in the control solution (black), at equilibrium block when applying 0.3 mM TEA (**A**), 14 nM ChTx (**B**), or 10 mM TEA (**C**) as a positive control (red) for respective ion channel or the toxin CboK7 (purple, 100 nM for Kv1.1 (**A**) and Kv1.3 (**C**), and 1 nM for Kv1.2 (**B**)). (**D**–**F**) Effect of CboK1 to CboK7 toxins on Kv1.1, Kv1.2, and Kv1.3 ion channels. Bars with individual data points (empty circles) indicate the remaining current fraction (RCF = I/I_0_, where I_0_ = peak current in toxin-free solution and I = peak current in the presence of toxin at given concentration at equilibrium block, see Materials and Methods for detail) values obtained from individual records at 100 nM for Kv1.1 (**D**) and Kv1.3 (**F**), and 1 nM for Kv1.2 (**E**) except for CboK1 which was tested at 100 nM concentration. Error bars indicate ± SEM (n ≥ 3). Statistical differences between RCF value at given concentration of respective toxin and RCF value in control condition were compared via one-way ANOVA with post-hoc Dunnet’s test and marked with ns = not significant; **, *p* < 0.01; ***, *p* < 0.001.

**Figure 5 toxins-15-00506-f005:**
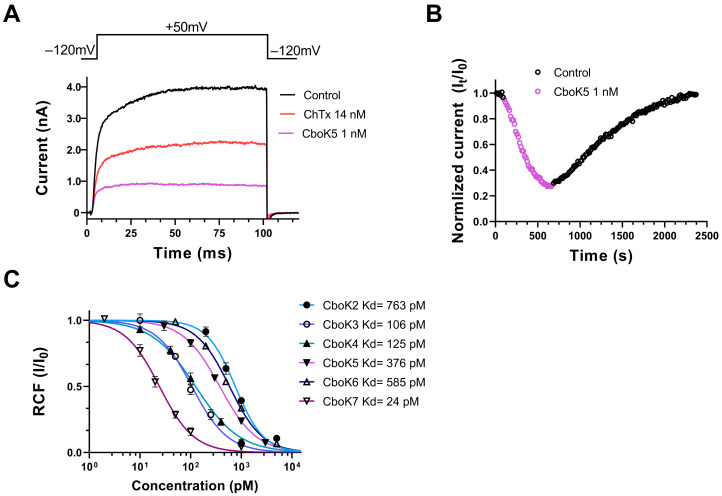
Effect of CboK peptide toxins on Kv1.2 currents. (**A**) Whole-cell currents were evoked in a CHO cell, transfected with the gene encoding Kv1.2, by applying 100 ms long depolarization pulses to +50 mV from a holding potential of −120 mV every 15 s. Current traces were recorded in the control solution (black), at equilibrium block by 14 nM ChTx as positive control (red) or by 1 nM CboK5 (purple). (**B**) Development of and recovery from the block of Kv1.2 current. Normalized peak currents (I_norm_ = I_t_/I_0_, see text) were plotted as a function of time. Data points in purple (empty circles) represent the application of 1 nM of CboK5 and upon reaching the equilibrium block, cells were perfused with toxin-free bath solution to determine the reversibility of inhibition (empty circle in black). (**C**) Concentration-dependent block of Kv1.2 by CboK peptides. The remaining current fractions (RCF = I/I_0_, where I_0_ = peak current in toxin-free solution and I = peak current in the presence of toxin at given concertation at equilibrium block) values were plotted against the toxin concentration and data points were fitted with the Hill equation (see Materials and Methods for detail). The best fit resulted in Kd values of 763 pM, 106 pM, 125 pM, 376 pM, 585 pM, and 24 pM for CboK2 to CboK7, respectively. Error bars indicate mean ± SEM and n ≥ 3.

**Figure 6 toxins-15-00506-f006:**
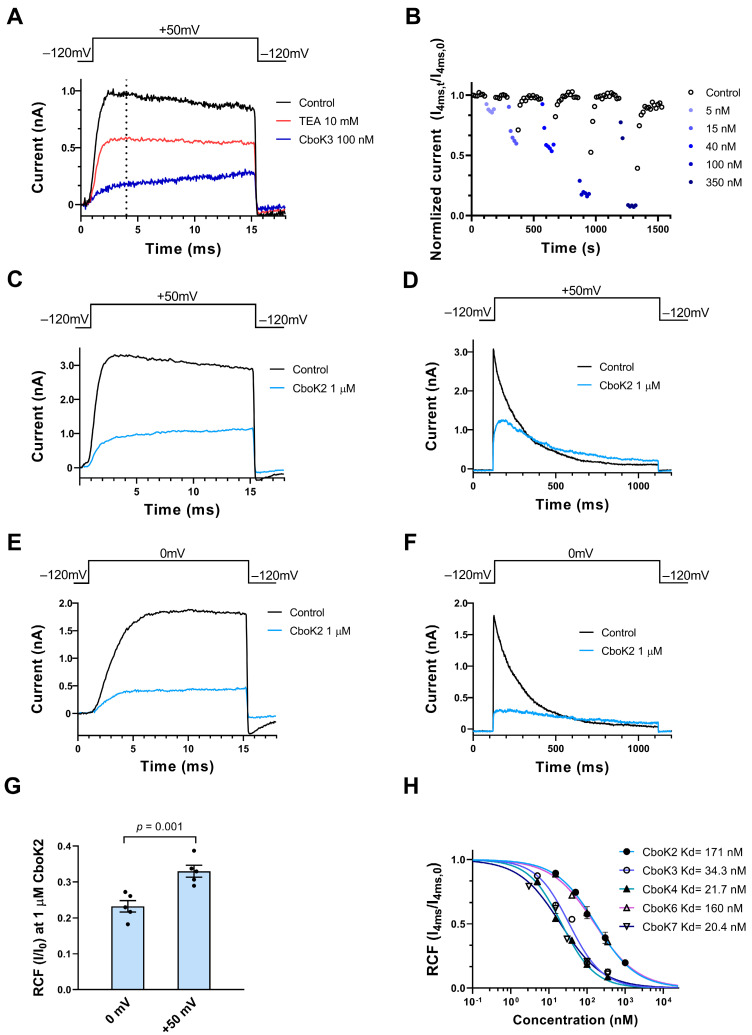
Inhibition of Kv1.3 currents by CboK peptides. (**A**) Whole-cell currents through Kv1.3 in a human T lymphocyte were evoked by applying 15-ms-long depolarization pulses to +50 mV from a holding potential of −120 mV every 15 s. Current traces were recorded in the control solution (black) and at equilibrium block by 10 mM TEA (red) or 100 nM of CboK3 (blue). The vertical dashed line indicates the isochronal time point (4 ms) at which Kv1.3 peak currents were taken. (**B**) Time course of development and removal of Kv1.3 current blockage by different concentrations of CboK3 peptide. Normalized currents were determined at the 4 ms time point (I_norm_ = I_4ms,t_/I_4ms,0_, see text, isochronal RCF) and plotted against time. Data points in shades of blue (filled circles) represent the application of CboK3 (5−350 nM) to the cell. After reaching the inhibition equilibrium at each concentration of toxin, cells were perfused with the control solution (lacking the toxin) to display the reversibility of the block (data points in black, empty circles). (**C**–**F**) The voltage-dependent inhibition of the Kv1.3 current by 1 µM of CboK2. The T cell was depolarized by applying voltage steps of +50 mV (**C**,**D**) or 0 mV (**E**,**F**) from −120 mV holding potential for 15 ms or 1 s long durations as indicated above the panel, every 15 s (for 15-ms-long depolarizations) or every 60 s (for 1-s-long depolarizations). The current traces were recorded in the presence of external solution (black traces) or at equilibrium block in the presence of 1 µM CboK2 (light blue traces). (**G**) Statistical analysis of RCF (I/I_0_) values obtained during 15-ms-long depolarization at 0 mV (panel **E**) and +50 mV (panel **C**). The filled black circles indicate individual data points. Error bars represent the mean ± SEM (n = 5). The *p* value from paired *t*-test is given above the bar plots. (**H**) Concentration-dependent inhibition of Kv1.3 by CboK peptides. Isochronal RCF (I_4ms_/I_4ms,0_) values were plotted as a function of toxin concentration and data points were fitted with the Hill equation (see Materials and Methods). Error bars indicate mean ± SEM and n ≥ 3. The best fits resulted in Kd values as indicated.

**Table 1 toxins-15-00506-t001:** Amino acid sequences and MW of isolated peptide toxins from subfraction F-II-13.

Peptide Name	Retention Time (min)	Amino Acid Sequence	Length	MW (Da)
CboK1	18.4	NVACVHRTCDSNCKRNGYKSGKCINRKCNCYPH	33	3765.1
CboK2	20.9	TVCVYRTCDKDCKRRGYRSGKCINNACKCYPY	32	3760.4
CboK3	21.7	TIINVKCTSPKQCLKPCKDLYGPHAGEKCMNGKCKCYKV	39	4323.5
CboK4	22.3	TFINVKCTSPKQCLKPCKDLYGPHAGEKCMNGKCKCYKP	39	4357.9
CboK5	22.9	IFINVKCSLPQQCLRPCKDRFGQHAGGKCINGKCKCYP	38	4249.8
CboK6	23.2	TTINVKCSLPQQCLRPCKDRFGQHAGGKCINGKCKCYP	38	4191.3
CboK7	25.3	TFINVKCTSPKQCLKPCKDLYGPHAGEKCMNGKCKCYKV	39	4298.5

**Table 2 toxins-15-00506-t002:** Comparison of Kd values of CboK peptides with their closest relatives.

Toxin Name	Accession Number	Systematic Name	Sequence	% Identity	Kd Values		
					Kv1.1	Kv1.2	Kv1.3
**A**							
**CboK1**	**C0HM73**	**α-KTx 10.5**	** NVACVHRTCDSNCKRNGYKSGKCINRKCNCYPH- **	**100%**	*****	*****	*****
**Cotx1**	O46028	α-KTx 10.1	-AV..Y....KD...R..R......NA.K...Y-	68%	24.4 µM	27 nM	5.3 µM
**II-10.4**	C0HJW2	α-KTx 10.4	-AV..Y....KD...R..R......NA.K...YA	66%	*	3.6 nM	72 nM
**B**							
**CboK2**	**C0HM75**	**α-KTx 10.6**	** TVCVYRTCDKDCKRRGYRSGKCINNACKCYPY- **	**100%**	*****	**760 pM**	**171 nM**
**Cotx1**	O46028	α-KTx 10.1	A...............................-	97%	24.4 µM	27 nM	5.3 µM
**II-10.4**	C0HJW2	α-KTx 10.4	A...............................A	94%	*	3.6 nM	72 nM
**CoTx2**	P58504	α-KTx 10.2	VA..........TS.K................-	84%	1 µM	*	*
**C**							
**CboK3**	**C0HM77**	**α-KTx 2.22**	** TIINVKCTSPKQCLKPCKDLYGPHAGEKCMNGKCKCYKV **	**100%**	**1 µM**	**106 pM**	**34.3 nM**
**CboK4**	**C0HM72**	**α-KTx 2.23**	** .F....................................P **	**95%**	**1.5 µM**	**125 pM**	**21.7 nM**
**CboK7**	**C0HM78**	**α-KTx 2.24**	** .F..................................... **	**97%**	**141 nM**	**24 pM**	**20.4 nM**
**II-12.5**	C0HJW6	α-KTx 2.16	......................................I	97%	*	700 pM	26.2 nM
**Ce4**	P0C164	α-KTx 2.11	..............L...EI..I...A...........I	85%	ND	ND	0.98 nM
**II-12.8**	C0HJW5	α-KTx 2.17	..............L...QI......A.......H.S.I	82%	4.8 nM	2.9 nM	*
**D**							
**CboK5**	**C0HM74**	**α-KTx 2.10**	** IFINVKCSLPQQCLRPCKDRFGQHAGGKCINGKCKCYP **	**100%**	*****	**376 pM**	**126 nM**
**Ce3**	P0C163	α-KTx 2.10	......................................	100%	ND	ND	366 nM
**E**							
**CboK6**	**C0HM76**	**α-KTx 2.21**	** TTINVKCSLPQQCLRPCKDRFGQHAGGKCINGKCKCYP **	**100%**	*****	**585 pM**	**160 nM**
**Ce3**	P0C163	α-KTx 2.10	IF....................................	95%	ND	*	366 nM
**II-10.9**	P45629	α-KTx 2.3	I......TS.............................	92%	*	*	*
**II-10.5**	C0HJW1	α-KTx 2.15	IF......S.....K...KA..................	84%	*	300 pM	8.3 nM

Sections in the table contain the peptide common names, UniProt accession numbers, systematic names, and the aligned amino acid sequence of peptides with high sequence similarity to CboK1 (A), CboK2 (B), CboK3, CboK4 and CboK7 (C), CboK5 (D), and CboK6 (E), and their Kd values measured on Kv1.1 to Kv1.3 ion channels (highlighted in bold). In the “sequence” column, identical positions to CboK toxins are indicated by dots whereas (-) represents the gaps. In the “Kd values” column, data in bold are from this study, other data are from literature (see citations in the corresponding sections of the discussion). Missing values indicate that either the peptide does not inhibit that specific channel (*) or its activity is not determined (ND).

## Data Availability

The raw data supporting the conclusions of this article will be made available by the authors, without undue reservation.

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
