# Peer review of "Of Seven New K+ Channel Inhibitor Peptides of Centruroides bonito, α-KTx 2.24 Has a Picomolar Affinity for Kv1.2"

_toxins, 2023, doi:10.3390/toxins15080506_

Round 1
Reviewer 1 Report
This manuscript clearly and concisely presents the isolation, sequencing, and electrophysiological investigation of seven scorpion toxins. It might be published in Toxins journal after several revisions.
Major comment:
In the section 2.1 it would be better to explain, why F-II, but not F-I or F-III, was used for further investigation.
P2 L92 and P15 L475
The carboxymethylated and carboxylated peptides have been mentioned but it seems that no modifications have been made except for reduction and alkylation. Is it true?
P7 Figure 4
On figures 4D, E, F please add information about the statistically significance of differences in ionic currents between the control and toxin conditions marked as p < 0.1; * p < 0.01; etc. And add corresponding text in the methods section. In addition, I believe that "100 nM" should be deleted from figure 4E (first bar).
Minor suggestions:
Page 1 Lines 10 and 12
In the abstract the word "whereas" was used twice per sentence. Authors could replace one of them.
P1 L15
In the sentence "CboK2 and CboK3 inhibited >10% and CboK7 inhibited ~42% 15 of Kv1.1 currents at 100 nM concentration" ">" and "~" could be replaced to "more" and "about".
P2 L78
"2.1. . Isolation and Sequence determination of CboK peptides" Remove extra dot in headline.
P3 Table 1
What is the point to specify peak identifier? Its first part "F-II-13" is identical for all peptides and it is not clear where the numbers "18.4" or " 20.9" come from. You could replace this column to UniProt ID or systematic names.
P4 L121 and L125
Please replase " amino acid identity " to " amino acid sequence identity ".
Specify (in particular on figure 4) that you use human channels or remove "hKv1.2" (P8 L241) from the caption of the figure 5, leaving this information only in the methods section.
Authors may combine figure 7 with figure 6.
P15 L499
Please change "1x10−10" to "1x10−10". Add superscript.
P16 L522-523
Delete "cell", it was used twice.
Author Response
First of all, we would like to Thank Reviewer 1 the time and efforts devoted to the review of our manuscript. We are also thankful for the recommendations. We have addressed them as follows:
Major comments:
(1) In section 2.1 it would be better to explain, why F-II, but not F-I or F-III, was used for further investigation.
Answer: Thanks for your comment. This is based on our experience as described previously by Possani et al., [1]. Fraction II of Sephadex G-50 contains all toxins that affect ion-channels, Fraction I contains higher molecular weight components, mainly enzymes, whereas F-III contains components of low molecular weights, none of which have so far being described as specific for K-channels.
We have inserted the explanation in manuscript.
(2) P2 L92 and P15 L475
The carboxymethylated and carboxylated peptides have been mentioned but it seems that no modifications have been made except for reduction and alkylation. Is it true?
Answer: Yes, you are right, the only procedure used was to reduce and alkylated the peptide, and the reagents and solvents were cleaned by passing the sample by HPLC. Thus, no other modifications were made.
(3) P7 Figure 4
On figures 4D, E, F please add information about the statistically significance of differences in ionic currents between the control and toxin conditions marked as p < 0.1; * p < 0.01; etc. And add corresponding text in the methods section. In addition, I believe that "100 nM" should be deleted from figure 4E (first bar).
Answer: Thank you for your suggestion to highlight the statistical significance between the RCF value in control and upon toxin exposure. We have indicated the statistically significant value as suggested by the Reviewer in Figure 4, panel D, E and F. We have also included the description about statistical methods in “Materials and Methods” section. We have also removed the “100 nM” in the bar graph from Figure 4E.
Minor suggestions:
Page 1 Lines 10 and 12
In the abstract the word "whereas" was used twice per sentence. Authors could replace one of them.
Answer: Thanks, we have corrected it.
P1 L15
In the sentence "CboK2 and CboK3 inhibited >10% and CboK7 inhibited ~42% 15 of Kv1.1 currents at 100 nM concentration" ">" and "~" could be replaced to "more" and "about".
Answer: Thanks for the suggestion we have modified it. Furthermore, we have made an error in the abstract that is now corrected, which we spotted when we made the recommended modifications.
P2 L78
"2.1. . Isolation and Sequence determination of CboK peptides" Remove extra dot in headline.
Answer: We have removed it.
P3 Table 1
What is the point to specify peak identifier? Its first part "F-II-13" is identical for all peptides and it is not clear where the numbers "18.4" or " 20.9" come from. You could replace this column to UniProt ID or systematic names.
Answer: The peak identifier represents the elution time of the respective peak from RP-HPLC column. Reviewer is right, the column name was a bit confusing. Therefore, now we have replaced this column with the retention time points of respective toxins in HPLC chromatogram.
We hope that now it is clear and more meaningful.
Thank you for pointing out this issue.
P4 L121 and L125
Please replace " amino acid identity " to " amino acid sequence identity ".
Answer: We have replaced it. Thanks
Specify (in particular on figure 4) that you use human channels or remove "hKv1.2" (P8 L241) from the caption of the figure 5, leaving this information only in the methods section.
Answer: We have removed the “h” from “hKv1.2”
Authors may combine figure 7 with figure 6.
Answer: Thanks for the great suggestion. We have merged figure 7 into figure 6H and also modified the figure reference in text.
P15 L499
Please change "1x10−10" to "1x10−10". Add superscript.
Answer: We have modified it. Thanks
P16 L522-523
Delete "cell", it was used twice.
Answer: We have corrected it.
References
- Possani, L. D., Becerril, B., Delepierre, M., & Tytgat, J. (1999). Scorpion toxins specific for Na+‐European journal of biochemistry, 264(2), 287-300.
Finally, we hope that Reviewer 1 will find the revised manuscript acceptable in Toxins.
Reviewer 2 Report
The authors present a remarkably comprehensive investigation of the isolation of 7 new ion channel blocking peptides obtained from the recently discovered scorpion, Centruroides bonito. The rationale for the investigation was that, in general, unique peptides that block specific ion channels may have a niche in treating some newly discovered neurological disorders, and that perhaps in the future there will be a way to deliver these peptides past the blood brain barrier to treat patients suffering from epilepsy.
The Introduction was detailed and informative, setting up the rationale and goals of the investigation. In the Results section, the basic isolation and molecular characterizations of the peptides was clearly presented, with figures 1-2 and table 2 being very informative. While not expected, the phylogenic classifications and lineage of the peptides presented in figure 3 was very informative.
In the next section of Results, the pharmacological characterization of the inhibition exerted by the peptides on relevant ion channels was presented. Using classic patch clamp methods with CHO cells transfected with the channel of interest, the authors demonstrate typical conductance, conductance inhibited by known inhibitors of the channel tested, and then a comparison of the inhibition of the channel with various concentrations of each venom derived peptide. Figure 4 displays these data very well. The remainder of the results present in great detail the specificity of key peptides to inhibit ion channels of interest that are associated with the diseases alluded to in Introduction and again mentioned in Discussion.
The data support the concepts put forth by the authors in Discussion, with the comparison of other peptides in venom obtained from different species of scorpion displayed in table 2 being of interest. Overall, the work is comprehensive, detailed, uses methods that are well described, and provides information of interest to investigators involved with channelopathies. I have no important criticisms.
Author Response
We would like to thank Reviewer 2 the time and efforts devoted to the review of our manuscript. We were very happy to read the positive review, we were delighted to read that Reviewer 2 found our manuscript suitable for publication in Toxins.